# Preliminary Analysis of Relationships between COVID19 and Climate, Morphology, and Urbanization in the Lombardy Region (Northern Italy)

**DOI:** 10.3390/ijerph17196955

**Published:** 2020-09-23

**Authors:** Massimiliano Fazzini, Claudia Baresi, Carlo Bisci, Claudio Bna, Alessandro Cecili, Andrea Giuliacci, Sonia Illuminati, Fabrizio Pregliasco, Enrico Miccadei

**Affiliations:** 1INGEO, University of Chieti-Pescara, 66100 Chieti, Italy; enrico.miccadei@unich.it; 2Poliambulanza Foundation Hospital, 25124 Brescia, Italy; claudia.baresi@poliambulanza.it (C.B.); claudio.bna@poliambulanza.it (C.B.); sonia.illuminati@poliambulanza.it (S.I.); 3School of Science and Technology, University of Camerino, 62032 Camerino, Italy; carlo.bisci@unicam.it; 4Department of Geology, University Roma Tre, 00187 Roma, Italy; cecili@uniroma3.it; 5Department of Physics, University of Milan Bicocca, 20124 Milan, Italy; andrea.giuliacci@gmail.com; 6IRCCS Istituto Ortopedico Galeazzi, University of Milan, 20136 Milan, Italy; fabrizio.pregliasco@unimi.it

**Keywords:** Coronavirus disease 2019 (COVID-19), temperate sub-continental climate, Lombardy, temperature, solar radiation, Italy

## Abstract

The coronavirus disease 2019 (COVID-19) pandemic is the most severe global health and socioeconomic crisis of our time, and represents the greatest challenge faced by the world since the end of the Second World War. The academic literature indicates that climatic features, specifically temperature and absolute humidity, are very important factors affecting infectious pulmonary disease epidemics - such as severe acute respiratory syndrome (SARS) and Middle East respiratory syndrome (MERS); however, the influence of climatic parameters on COVID-19 remains extremely controversial. The goal of this study is to individuate relationships between several climate parameters (temperature, relative humidity, accumulated precipitation, solar radiation, evaporation, and wind direction and intensity), local morphological parameters, and new daily positive swabs for COVID-19, which represents the only parameter that can be statistically used to quantify the pandemic. The daily deaths parameter was not considered, because it is not reliable, due to frequent administrative errors. Daily data on meteorological conditions and new cases of COVID-19 were collected for the Lombardy Region (Northern Italy) from 1 March, 2020 to 20 April, 2020. This region exhibited the largest rate of official deaths in the world, with a value of approximately 1700 per million on 30 June 2020. Moreover, the apparent lethality was approximately 17% in this area, mainly due to the considerable housing density and the extensive presence of industrial and craft areas. Both the Mann–Kendall test and multivariate statistical analysis showed that none of the considered climatic variables exhibited statistically significant relationships with the epidemiological evolution of COVID-19, at least during spring months in temperate subcontinental climate areas, with the exception of solar radiation, which was directly related and showed an otherwise low explained variability of approximately 20%. Furthermore, the average temperatures of two highly representative meteorological stations of Molise and Lucania (Southern Italy), the most weakly affected by the pandemic, were approximately 1.5 °C lower than those in Bergamo and Brescia (Lombardy), again confirming that a significant relationship between the increase in temperature and decrease in virulence from COVID-19 is not evident, at least in Italy.

## 1. Introduction

During the second half of December 2019, the World Health Organization (WHO) reportedly received information about an epidemic with unidentified etiology from Wuhan (Hubei, China). On 11 February 2020, this epidemic was officially named coronavirus disease 2019 (COVID-19) and was acknowledged as an infectious disease resulting in a public health emergency, as it quickly spread within China and to 24 additional countries throughout the world [1,2].

Shortly thereafter, the SARS-CoV-2 or COVID-19 pandemic had become a major global health threat. The last time the world responded to a global emerging disease epidemic of a similar scale with no access to vaccines was in 1918–1919, with the H1N1 “Spanish” influenza pandemic. While our understanding of infectious diseases and their prevention is now very different than that in 1918, most of the countries across the world face the same challenge today with COVID-19, a virus with comparable lethality to H1N1.

Coronavirus is a specific type of virus affecting the respiratory tract. It causes several illnesses, ranging from colds and pneumonia to severe acute respiratory syndrome (SARS) [3,4].

On the night of 20 February 2020, the first Italian case of COVID-19 was confirmed in Codogno, in the Lombardy Region. During the following week, Lombardy experienced a very rapid increase in the number of infections. At the time of detection of the first COVID-19 case, the epidemic had already spread in most municipalities of southern Lombardy [5].

During the early stages of the COVID-19 epidemic in Lombardy, we observed the formation of three major clusters identified around the cities of Codogno, Bergamo, and Cremona. Later, the epidemic started to spread throughout the region and, subsequently, all over Italy within a short time, albeit with a minor spread due to the lockdown that occurred earlier in the southern than in the northern regions. Similar quick spreads of COVID-19 in France, Germany, Spain, USA, and South Korea, and subordinately to more than one hundred countries, causing thousands of deaths, led the WHO to declare it as pandemic [6] on 11 March 2020.

As of 31 May 2020, the total number of cases reported by the authorities reached 233 thousand, and approximately 32 thousand of these cases resulted in death. Northern Italy was mostly hit, and the region with the highest number of cases was Lombardy, which registered approximately 88 thousand cases, with a percentage of infected equal to 0.86% of the regional population and 15.3 thousand victims (Figure 1). The neighboring regions of Northern Italy followed in the list of regions with the highest rate of infections. The virus mostly impacted individuals older than 50 years in Italy.

As of 20 May, Italy had the sixth-highest number of coronavirus cases after the United States, Russia, Brazil, the United Kingdom, and Spain [7,8,9].

The present preliminary study aims to individuate possible correlations of COVID19 magnitude with environmental and social-economic features, adopting simple statistical methods.

## 2. Geography and Climatology of Lombardy

Lombardy is one of the twenty administrative regions of Italy and is located in the northwest of the country, with an area of 23,844 square kilometers (9206 square miles). Approximately 10.5 million people live in Lombardy, representing slightly more than one-sixth of Italy’s population, and 18% of Italy’s GDP is produced in the region, making it the most populous, richest, and most productive region in the country. Lombardy is also one of the top regions in Europe according to the same criteria.

Milan’s metropolitan area is the third largest in Italy after Naples and Rome and the third most populated functional urban area in the EU. Lombardy has a wide array of climates due to local variances in elevation, proximity to large inland water, and large metropolitan areas (e.g., Milan).

The climate of the region (Figure 2) is mainly humid subtropical (cf. according to Köppen and Geiger, 1954 [10]), but it is truly humid subcontinental, especially along the Po plain and the largest valley floors. 

However, the area exhibits significant variations with respect to the Köppen model, especially for the winter season, which is normally long, damp, and cold in Lombardy, with several days without daytime thaw. In addition, there are relevant seasonal temperature variations: In Milan, for the 1981–2010 timespan, the average temperature was 2.7 °C (36.9 °F) in January and 24.5 °C (77 °F) in July. The annual average temperature (AAT) is 13.1 °C (55.5 °F). A peculiarity of the regional climate is the thick fog and, consequently, the smog that frequently covers the Po plains between October and February.

Along the Alpine foothill area, characterized by an oceanic climate (Köppen Cfb), numerous lakes (Maggiore, Como, Lugano, Iseo, and Garda) locally exert a mitigating influence—just to 15 °C of the AAT along their coasts—allowing the cultivation of typical Mediterranean crops (olives, orange, and citrus fruit). In the medium-high hills and mountains, the climate becomes humid continental, with cold winters and mild to fresh summers (Köppen Dfb).

Along the alpine valleys, the climate is relatively mild but with frequent thermal inversion, while it can be severely cold at elevations above around 1500 m, with abundant snowfalls (627 cm/season of fresh snow at Passo Tonale, 1883 m a.s.l.). Above 3000 m, large glaciers are present in the Adamello-Presanella, Ortles-Cevedale, and Bernina massifs.

Precipitation is more abundant in the pre-alpine area, ranging from 1500 to 2200 mm (59.1 to 80.7 in) annually, but is also abundant in the plains and alpine areas, with an average of 600 to 850 mm (23.6 to 33.5 in) annually; approximately 1013 mm (39.9 in.) of precipitation falls annually in Milan. The total annual rainfall in the region is, on average, 853 mm. The pluviometric regime is of “padan” (i.e. typical of the River Po plain) type [11] in the lower Po plain, slightly bimodal, with an absolute minimum in winter, which is common to all regional precipitation regimes. The second lowest values occur in July, and two maxima occur in November and April.

In the highland area, in the hills, in the pre-alpine sector, and in the Orobie Alps, the regime becomes pre-alpine-subalpine, with two similar maxima in spring and autumn and two moderate minima in the solstice seasons. Along the Alps, the regime becomes subcontinental and continental, unimodal, with a summer maximum and a moderate snowy minimum in winter.

### Meteorological Analysis for the Period 1 March–20 April

From a meteorological point of view, the period under study showed a clear discontinuity with respect to the first two months of the year, which were dominated by prevailing anticyclone conditions and with an extremely mild thermal climate (up to 4 °C above the climatic average). During the first part of March, four fronts passed through the Atlantic in rapid succession, which brought significant rainfall in Northern Italy. The most intense of these fronts occurred between 2 and 3 March, accompanied by intense atmospheric phenomena and abundant snowfall even below 1000 m. Temperatures fell below the average climate conditions by approximately 2 °C. Between days 11 and 21 March, it was characterized by more stable meteorological conditions due to the increased presence of the Azores anticyclone. During this phase characterized by dry and mild weather, the passage of a single perturbation was experienced, with modest effects on day 14th. In its wake, however, there was a significant thermal decrease due to the cold continental polar currents crossing the Po Valley.

This cold phase coincided with the period of maximum epidemiological expansion of coronavirus in Lombardy. The last part of March began with a late burst of Arctic marine air masses over the country. Between 22 and 23 March, the development of a blocking configuration in northern Europe favored the transport of north-eastern currents up to the Mediterranean, with a sharp and generalized decrease in temperatures, even in the presence of very low rainfall, with snow up to high plains (approximately 300 m). In the following days, the weather remained variable to improve significantly between 28 and 29 March due to the weakening of the low pressure over the Mediterranean, while the passage of a new Atlantic cold front characterized the last two days of the month, with abundant rainfall on the 30th day.

The first two weeks of April were characterized by a long phase of stable and dry conditions, with very mild temperatures (1–2 °C above the climate average) due to the persistent presence of a subtropical anticyclone that extended over a large part of southern Europe. This long phase, dominated by weather stability, was temporarily interrupted only on 14 April, when Lombardy was rapidly crossed by a cold front coming from northern Europe and causing weak rainfalls. Cold currents descending from northern Europe also favored a sharp thermal decline in much of the country. Between 15 and 18 April, a new phase of fair weather occurred due to a high-pressure system that elongated from North Africa to the Alps. On the 19th, the approach of low pressure from the western Mediterranean caused the first modest worsening of the weather, resulting in significant precipitation on the 20th. Despite the worsening of the weather, due to the lukewarm air mass following the disturbance, the temperatures remained mild and higher than the average climate.

## 3. Material and Methods

This preliminary study only considers climatological parameters, even though they have extremely short temporal steps (hourly data) in the first phase of the research. Parameters related to air quality were also considered, and the response of the epidemiology was analyzed with respect to some important parameters, such as sulfur dioxides, nitrogen dioxides, ozone, and total particulate matter. However, after discussion with atmospheric physicists, it was evident that these parameters vary extremely rapidly as they move away from their main production sources, and therefore, the results could have been misleading.

The period analyzed by the study was forced to cover the time span from 1 March to 20 April as the provincial ATS provided daily epidemiological data for each municipality during only this period. The analysis of temperatures and precipitation extended to only 30 April, while the study is still in progress for the city of Brescia, as daily data from the Poliambulanza Foundation Hospital, the second largest in the city, are available for this city. The daily municipal epidemiological parameters used for the analysis were provided by the provincial ATS (territorial health company), while the meteorological and climatic parameters—minimum and maximum temperature (°C), average relative humidity (%), diurnal solar radiation sum (W/m^2^), wind direction and speed (azimuth angle and m/s, respectively), and evaporation (mm)—were provided by automatic weather stations (AWSs) owned by the ARPA Lombardy regional meteorological service. Table 1 and Figure 3 show the geographical characteristics of the considered AWSs that were highly representative for the most important pandemic outbreaks.

Some morphological and morphometric features were also considered in the analysis at each AWT—distance from the Pre-Alps, width, and main direction of the valley sector in which the AWT is located, distance from the Po River, elevation—and quantified using the ESRI ArcGIS platform.

As the studied area is very homogeneous from a climatological point of view (Table 2, Figure 4), a daily average was calculated for each of the climatic variables considered for the purpose of statistical analysis.

To determine the relationship between the evolution of the pandemic and climate characteristics at the mesoscale, all the available epidemiological parameters (number of swabs performed, number of hospitalizations in the emergency room, number of hospitalizations in the COVID-19 area, number of hospitalizations in intensive care, and deaths) were considered. However, the only epidemiological parameter not heavily impacted by significant daily mistakes due to administrative and legal reasons is the ratio between the first positive swabs and the total number of swabs collected per day (Figure 4). Therefore, only this parameter was considered for analysis purposes. Figure 4 highlights that no sound correlation exists between temperature and first positive swabs.

In accordance with the scientific literature (ECDC 2020 [12]), an average incubation period of 5 days was considered; thus, for example, for the statistical analysis, the number of first positive swabs reported on 15 April was compared to the meteorological data from 10 April.

As the data are not normally distributed, Mann–Kendall (MK) rank correlation, multi-regression, and stepwise forward analysis were utilized to examine the possible correlation between variables.

The former (Mann 1945 [13], Kendall 1975 [14], Gilbert 1987 [15]) aims to statistically assess if there is a monotonic upward or downward trend of the variable of interest over time, i.e., if the variable consistently increases or decreases through time, even though the trend may not be linear. The MK test can be used in place of a parametric linear regression analysis, which can be used to test if the slope of the estimated linear regression line is different from zero. The regression analysis requires that the residuals from the fitted regression line be normally distributed, an assumption not required by the MK test, that is, the MK test is a non-parametric (distribution-free) test.

The latter is a method of fitting regression models in which the choice of predictive variables is carried out by an automatic procedure (Hocking, 1976 [16]). In each step, a variable is considered for addition to or subtraction from the set of explanatory variables based on a sequence of F-tests.

Several previous studies highlighted some relationship between the pandemic and relative and absolute humidity, as well as maximum temperature, while it appears that no author has yet discussed relationships between the pandemic and the intensity of solar radiation.

Araújo and Naimi (2020 [17]) calculated an evident decrease in the virulence of COVID-19 throughout the world as the temperature increased to over 29 °C and relative humidity decreased below 40%. Scafetta (2020 [18]) highlighted that the 2020 winter season in the Wuhan region was extraordinarily similar to that in the northern Italian provinces of Milan, Brescia, and Bergamo, where, in March, the epidemy was devastating. This similarity may indicate that the activity of the coronavirus 19 is stronger with air temperatures between 4 °C and 11 °C. In the same area of China, Ma et al. (2020 [19]) and Liu et al. (2020 [20]) demonstrated that absolute humidity was negatively associated with the daily death counts related to COVID-19. Once more in China, Wang et al. (2020 [21]) found that high values of relative humidity have strong influence on the R-value, with a significance level of 1%. Specific and absolute humidity finally exhibited a significant and consistent distribution with the seasonal behavior of respiratory viruses at latitudes between 30° N and 50° N (Sajadi et al., 2020 [22]). This result almost coincides quantitatively with that calculated by Sahin (2020 [23]) and Sahin et al. (2020 [24]) for the Anatolian Peninsula, which has an average spring climate that is very similar to that of Lombardy.

Some authors did not detect any statistically significant relationship between the aforementioned climatic variables and COVID-19. Tosepu et al. (2020 [25]) posited that average temperature was significantly correlated with COVID-19. However, it was claimed that minimum and maximum temperature, rainfall, and humidity were not significantly correlated with the COVID-19 signal. Bashir et al. (2020 [2]) showed that average and minimum temperatures in New York City exhibited poor significance in the COVID-19 epidemic.

Prata et al. (2020 [26]) suggested that there is no evidence supporting that case counts of COVID-19 could decline if temperatures are above 25.8 °C.

Finally, Cheval et al. (2020 [27]) pointed out that local weather conditions of lowered temperature, mild diurnal temperature range, and low humidity may favor the transmission, while other studies claim there is no evidence that warmer weather can determine the decline in the case counts of COVID-19.

## 4. Discussion

The results of the Mann–Kendall test show in a peremptory manner that there is no climate variable that is acceptably correlated with the temporal evolution of the pandemic, with the exception of relative humidity RH% (value-0.548) with a level of significance of 1% (99% confidence intervals). The output of the stepwise forward multiple linear regression analysis (Andrews and Pregibon, 1978 [28]) showed that, considering the variables that explain at least 3% of the total variability, only three of them—relative humidity (RH%), solar radiation (Sol Rad) (directly related), and maximum temperature (Tmax, inversely correlated)—cover approximately 60.5% of the explained variability, with values of 35.9, 21, and 3.6%, respectively.

The multiple linear regression equation is:Y(first positive swabs) = 0.503 RH% + 0.03 SolarRadiance − 0.0582 Tmax

The percentages of the explained variability are considered satisfactory for the relative humidity and solar radiation—and in this case, they show that the spread of the pandemic is favored by increased sunshine and high relative humidity—while the statistical contribution provided by the maximum temperature is practically close to zero and, therefore, not significant.

Our study, relating only to the situation that occurred in Brescia and extended temporally until 30 June (Figure 5), shows that with the increase in average and, above all, maximum temperatures, a progressive decrease in virulence is recorded. Anyhow, it is also evident that after 20 April, the signal had become statistically unreliable, since “house-to-house” serological tests had started in the town, thus allowing the identification of many situations with clinically asymptomatic virus positivity. On the other hand, in the first part of July, in many areas with pronounced thermal seasonality (USA, Russia, Balkans) or with a very hot climate (e.g., Arabian Peninsula), the pandemic reached its peak, thus suggesting a statistically inverse relationship with temperatures in large cities and metropolises (source WHO).

Comparing population density, industrial activities (secondary), and services (tertiary) with the spatial distribution of COVID 19 in Lombardy, an extremely high correlation results; therefore, it would be obvious to look at them to individuate the main causes of the exceptional numbers that characterized the pandemic in this region. The data made available by ISTAT [29] on the number of activities belonging to the secondary and tertiary sectors were analyzed for the provinces of Milan, Bergamo, and Brescia (Table 3, and Figure 5 and Figure 6), comparing them with those relating to the province of Naples, the only in Italy having anthropic and industrial quantitative characteristics very similar to those of the three Lombardy provinces.

From the analysis of such data, unfortunately available only at the provincial scale, it is shown that population density is decidedly higher in the province of Naples than in the Lombardy provinces, but the ratio between COVID 19 cases and the number of inhabitants, and subordinately population density, is considerably lower in the southern province (about 0.09 vs. about 1; Table 3).

Regarding the density per square kilometer of the most important secondary sectors mainly favoring human contacts (Figure 6), it can be observed that in the province of Naples, manufacturing activities have a density slightly lower than that of the province of Milan, but they are much more widespread than in the provinces of Bergamo and Brescia.

For the tertiary sector (Figure 7), a similar signal is observed for retail and wholesale activities. Therefore, even these activities promoting close potential contact between people do not justify the widespread expansion of the virus in Lombardy.

The study area is characterized by a very sharp contrast between the large Po plain and the hilly and low mountainous reliefs of the pre-Alpine area, to the North. Some of the Covid19 hotspots are located along the bottom of NE-SW trending valleys crossing the latter area (Figure 8). The portion of the Po plain bordering the pre-Alps is characterized by the highest population density (up to 2200 inhabitants per sq. km between Bergamo and Brescia), as well as by the maximum concentration of industries. There, often the main built-up areas are located over low-angle alluvial fans or frontal moraines; both densities quickly decrease entering the side valleys, where only minor towns are located, such as Nembro—the town with the highest rate of COVID19-related victims (more than 14 deaths per 1000 inhabitants)—and Nuvolento, one of the towns less affected by the contagion (1 death among 2143 citizens). The same trend is self-evident moving to the South, toward the Po river, where agricultural practices dominate and most of the contagion seems to be related to local fairs.

For the future, it should be instrumental to understand whether there is a possible relationship between the temporal “rhythmicity” of coronavirus epidemics and the newly established global climate changes that are currently underway.

## 5. Conclusions

High temperatures do not seem to favor the decrease in COVID-19 cases; this is confirmed by the peremptory increase in the pandemic diffusion maxima after 10 May up to the first days of July in areas with a cold temperate climate (Köppen Df), such as Sweden and Denmark, where temperatures have become decidedly mild. The same epidemiological signal was also observed in areas with equatorial or monsoonal climates, where the thermal seasonality is absent or reduced, such as in the Indian and Nepalese territories (Köppen Am), and the Amazon and Peru (Köppen Af). The pre-monsoon period in these regions is notoriously the hottest of the year, with temperatures reaching 47 °C in the second half of May. Finally, in June and July, the pandemic reached the prolonged maximum in west and south USA, in Russia and in the Balkan area, as well as in the equatorial part of Brazil.

Finally, the countries with the highest percentage of infected people compared to the total population were those bordering the Persian Gulf, notoriously characterized by a hot desert climate (Köppen BH), with extremely high spring and summer thermal values (up to 52 °C) but with extremely low relative humidity (in Qatar at the end of June, a percentage of infected people of around 3.65% was reached).

A comparison between the maximum and average temperatures for the months of March and April, between the cities in Lombardy most affected by COVID-19 (Bergamo and Brescia) and two locations representative of the thermal climate of the southern regions (Molise and Basilicata) that were least affected by the pandemic, confirms the absence of statistical relationships between these two variables. In the AWS of Campochiaro (Molise) and Potenza (Basilicata), the temperatures were, on average, lower than those in the cities in Lombardy by approximately 1.5 °C, but there is no sign of a percentage increase in population with positive swabs.

For the city of Brescia, an ongoing study shows that very high maximum temperatures (above 29 °C) do not correspond to a drastic drop in the number of first positive swabs but rather to the statistically not significant oscillation of the parameter that, starting from 20 April, has shown a general trend toward a gradual, slow decline.

At the same time, the mesoscale morphologic analysis did not contribute to clarify the spatial distribution of the pandemic. For example, the lower Val Seriana (Nembro, in the Bergamo area), which had the highest percentage of deaths in the world in relation to the population, and the lower Chiese valley (Nuvolento, in the Brescia area), which was the least affected by the pandemic in the two provinces of Bergamo and Brescia, are arranged almost parallel to each other, being limited by reliefs with similar altitudes, and are located at similar distances (approximately 10 km) from the respective provincial chief towns (Figure 7).

Therefore, the high number of COVID-19 cases in Lombardy are most likely caused by the very high mobility of people deriving from the large number of secondary and tertiary activities occurring in all the “outbreak” areas of the epidemic. The population density is also very high (approximately 430 inhabitants per km^2^ and up to 2200 inhabitants per km^2^ in the foothill belt located between Codogno, Milan, and Brescia) which allows for the fast transmission of contagion. However, a comparison with the same parameters calculated for the province of Naples, where the pandemic was extremely less virulent, shows that even these parameters do not justify the presence of the large outbreak in Lombardy.

Possibly, the most important cause of the severe impact of the pandemic in the study area, considered that with the most severely polluted air in Europe (EEA, 2018 [30]), is the high concentration of particulate matter (PM, including PM10 and PM2.5) making the respiratory system more susceptible to infection and complications deriving from the coronavirus disease. This obviously increased the mortality rate for COVID19, but very likely also contributed to enhance the number of positive tests, making more severe the health condition of the infected.

Moreover, a similar effect could be related to the high average age per population of the two considered Provinces (ISTAT, 2020) [31].

Several other fundamental parameters (such as vehicular traffic, the use of personal protective equipment (PPE), and the most basic hygiene rules) are impossible to quantify for statistical analysis.

## Figures and Tables

**Figure 1 ijerph-17-06955-f001:**
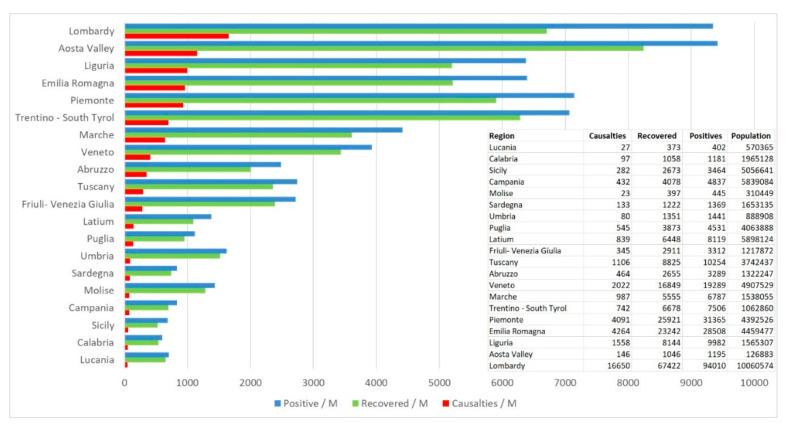
Regional distribution of coronavirus disease 2019 (COVID-19) on 1 July: Data per million population (source: National Civic Protection).

**Figure 2 ijerph-17-06955-f002:**
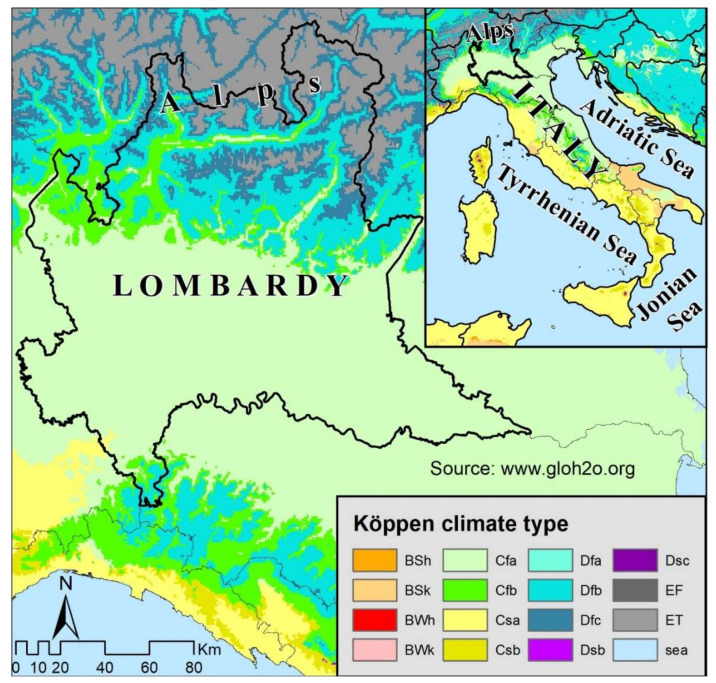
Climate map of Italy with evidenced Lombardy. For the codes, refer to Köppen-Geiger, (1954).

**Figure 3 ijerph-17-06955-f003:**
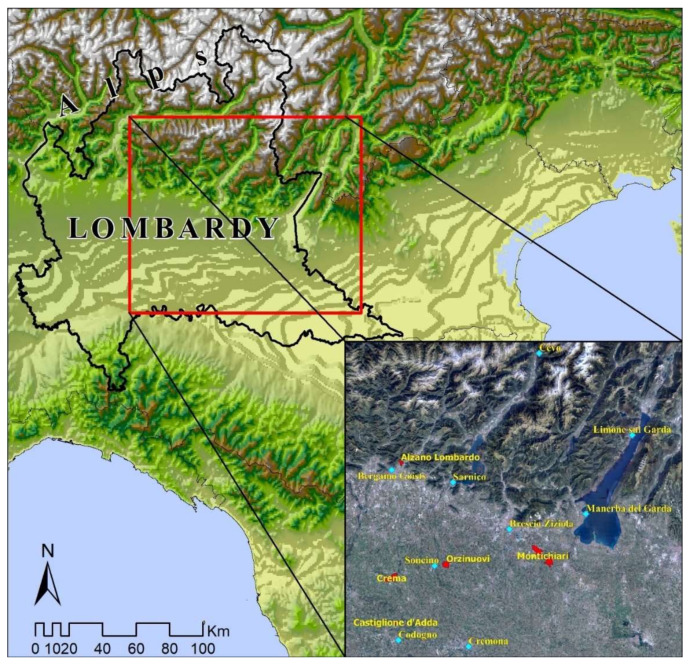
Map of Lombardy, with focus on the outbreak area: The weather stations used for the analysis are highlighted in light blue and the main epidemic outbreaks are highlighted in red.

**Figure 4 ijerph-17-06955-f004:**
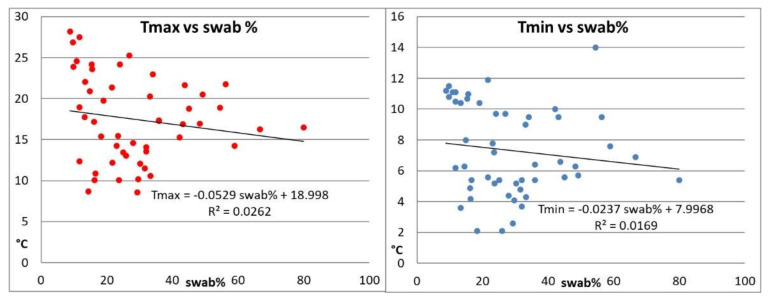
Relationships between percentage of first positive swabs and temperatures (maximum and minimum, to the left and to the right respectively) for the time span 1 March–20 April. R2 is the determination coefficient of the interpolating line.

**Figure 5 ijerph-17-06955-f005:**
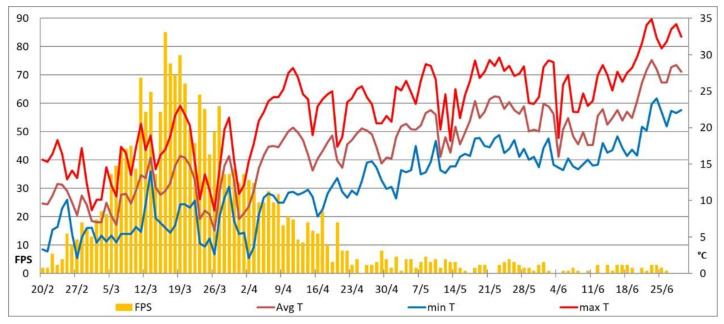
First positive buffers swabs (FPS) recorded in the Poliambulanza Institute (Brescia) and local thermal values for the period 1 April–30 June.

**Figure 6 ijerph-17-06955-f006:**
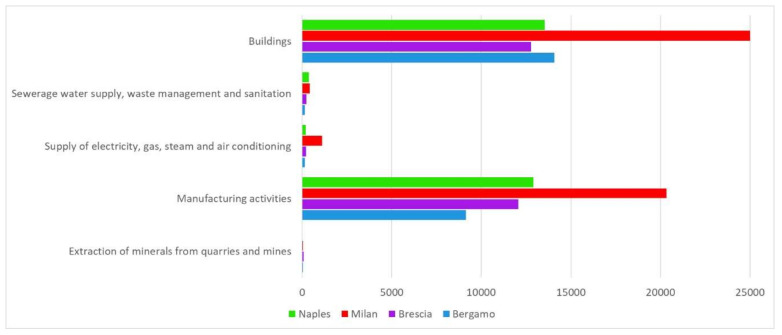
Presence of secondary sector in the four examined provinces.

**Figure 7 ijerph-17-06955-f007:**
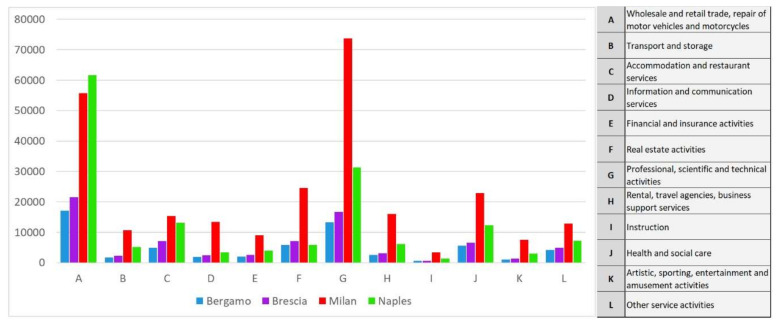
Number of firms operating in the tertiary sector in the four examined provinces.

**Figure 8 ijerph-17-06955-f008:**
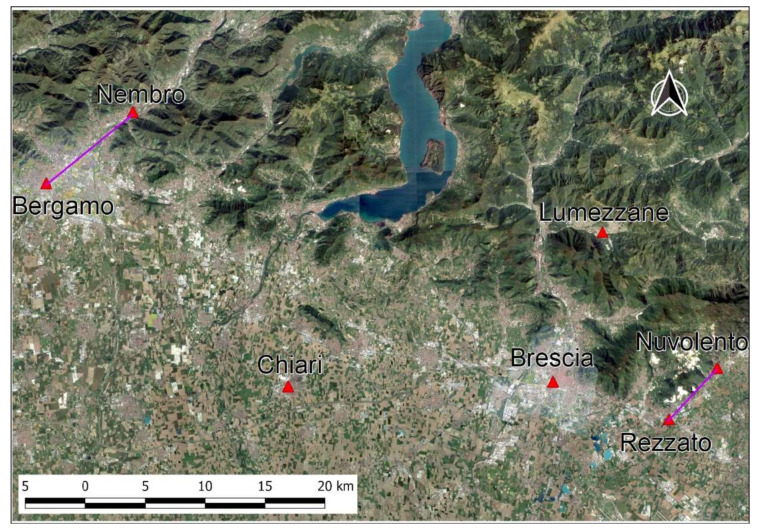
Location of the Seriana (Nembro) and Chiese (Nuvolento) valleys. The purple lines highlight that the two valleys are parallel to each other. Basemap Google Satellite.

**Table 1 ijerph-17-06955-t001:** Geographical features of the analyzed meteorological stations (AWSs).

AWS	Lat N	Long E	Elev
Bergamo Goisis	45.710°	9.680°	290 m
Brescia Zizola	45.513°	10.217°	70 m
Cevo	46.080°	10.369°	1128 m
Codogno	45.160°	9.706°	68 m
Cremona	45.137°	10.024°	43 m
Limone del Garda	45.907°	10.790°	43 m
Manerba del Garda	45.559°	10.570°	74 m
Sarnico	45.667°	9.963°	197 m
Soncino	45.399°	9.873°	87 m

**Table 2 ijerph-17-06955-t002:** Average climatological values for March and April 2020.

Variable		Bergamo	Brescia	Cevo	Codogno	Cremona	Limone	Manerba	Sarnico	Soncino
MARCH	avg T (°C)	8.5	9.1	4.1	9.2	8.7	10.2	9.5	9.0	9.3
	min T (°C)	4.3	4.8	0.9	4.6	3.5	7.0	6.8	5.3	4.5
	max T (°C)	13.4	13.5	8.2	14.1	13.8	14.5	12.4	13.4	14.1
	Precip. (mm)	88.4	43.8	103.8	47.8	31.4	63.2	58.0	85.4	50.4
	RH (%)	65.0	67.0	68.0	70.0	73.0	71.0	70.0	72.0	65.0
	Wind (m/s)	1.8	1.7	3.0	1.0	1.5	3.0	2.3	2.1	1.6
	Sol. rad. (W/m^2^)	503	498				511			
	Evapor. (mm)	3	4	0	5	4	6	5	5	6
APRIL	avg T (°C)	13.5	14.6	9.4	14.0	13.8	15.3	14.1	14.2	4.7
	min T (°C)	7.5	8.6	5.3	7.6	6.7	10.7	10.4	9.2	7.8
	max T (°C)	19.5	20.5	14.6	20.7	20.8	21.2	18.7	20.5	21.2
	Precip. (mm)	43.6	32.4	41.0	19.4	21.2	32.0	43.0	54.8	20.2
	RH (%)	51.0	53.0	66.0	58.0	56.0	67.0	68.0	63.0	59.0
	Wind (m/s)	2.0	2.2	3.1	1.9	2.0	4.2	3.3	3.1	2.0
	Sol. rad. (W/m^2^)	347	532				333			
	Evapor. (mm)	11	13	6	12	12	14	13	13	13

**Table 3 ijerph-17-06955-t003:** Main anthropic and pandemic features in some Italian Provinces.

Province	Surface (sq. km)	Million Inhabitants	Positive Cases	Positive/M People
Bergamo	2755	1.11	14,100	1.27
Brescia	4785	1.26	15,560	1.23
Milan	1575	2.35	24,080	1.02
Naples	1171	3.11	2654	0.09

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
