# Peer review of "Preliminary Analysis of Relationships between COVID19 and Climate, Morphology, and Urbanization in the Lombardy Region (Northern Italy)"

_ijerph, 2020, doi:10.3390/ijerph17196955_

Round 1

Reviewer 1 Report

The influence of climatic parameters on COVID-19 remains extremely controversial. The goal of this study is to quantify the existing relationship between several daily climate parameters (temperature, relative humidity, accumulated precipitation, solar radiation, wind direction and intensity, and evaporation), local morphological parameters, and new daily positive swabs for COVID-19, which represents the only parameter that can be statistically used to quantify the pandemic. The authors confirmed that a significant relationship between the increase in temperature and decrease in virology from COVID-19 was not evident, at least in the Italian peninsula. This study plays a positive role in better understanding the relationship between climate features and COVID-19. I believe readers will be very interested in the results of this paper. However, the method of the paper is not very clear. There are some major concerns in the paper that need to be revised.

My major concerns as follows:

  1. As a research paper, the introduction should not only introduce how the virus spreads, but also explain the disputes between climate features and disease epidemics mentioned in the abstract. List which scholars think that the spread of the virus is related to climate and which scholars think it has nothing to do with it.
  2. Line 266-268, could you please give us more details about methods? Such as equations of correlation and multi-regression analysis or references.
  3. Line 271-286, I think this part belongs to Method. The results should show correlation and regression results not the regression equation.
  4. Line 288-329 is literature review belongs to Introduction. The Section 6 belongs to Discussion.
  5. Generally speaking, it is not suitable to put pictures in the conclusion, so you can put this part in the discussion.

Minor concerns as follows:

  1. Line 145, Figure 2. Please add the scale and north compass
  2. Line 241, Figure 3. Please add the legend to illustrate the meaning of the different colors. The picture on the right is not very clear.
  3. Line 255, Table 2. The format of the Tables need to be adjusted to the requirements specified in the journal template. The header is at the top of the table.
  4. Line 263, What does " 5,3 days " mean?
  5. Figures 5 and 6 are not clear, so it is recommended to redraw them. The background can be white.

Reviewer 2 Report

The article is written very chaotically.

  1. Not all analyzed analysis stories are summarized. For example, population density, industrial activities, services, altitude, morphology are ignored.
  2. The title is too long. I propose to shorten it to one sentence.
  3. Use the term urbanization or human activity. Uniformly throughout the text.
  4. Are such short metrological periods sufficient to perform statistical tests? Observed meteorological conditions take place before the test, in various meteorological conditions.
  5. When translating the correlation of COVID-19 infections from meteorological conditions, one should take data from periods of different specificity (developed differentiation of daily values). I propose to complete the months of meteorological data with observations (after April).
  6. Statistical correlation analysis neglect population, industrial and service activities, morphological data (all measures in the article).
  7. For discussion, touch point 6, not point 5.
  8. In the process there is no comparison to human activity and morphology.
  9. The summary part does not refer to the research results presented in the article. It describes observations from other countries. The conclusions of the research are not sufficient.
  10. In conclusion the terms Val Seriana (Nembro) and the Chiese valley (Nuvolento), but these locations have not been found before.
  11. Figure 2, Figure 3 are of very poor quality.
  12. The titles of figures and tables should be written in capital letters.
  13. Follow the same pattern in tables 1, 2, 3.
  14. About rganize data in Table 2.
  15. Correct the drawing number (Figure 5 is or or).
  16. In figures 5 and 6, please provide earlier graphical forms for examining drawings (deletion of the color command, entering full names on the X axis, entering units on the Y axis).

17. Figure 7 and the reference text for it, I recommend reading the analytical part.

Author Response

xPlease see the attachment

Reviewer 3 Report

Dear authors,

Is there a relationship between climate, morphology and urbanization and COVID19? Preliminary analysis of environmental and pandemic data in the Lombardy region (Northern Italy)By Fazzini et al.

General comments

Topic:is relevant for publishing in International Journal of Environmental Research and Public Health.

Introduction:The introduction section is complete with applicable background information about COVID19 in the Lombardy region in the north of Italy. However, the aim of study did not express here!

Geography and climatology of Lombardy: It gives a brief but enough description of geography and climate conditions of Lombardy. The aim of study is not also here.

Material and Methods:is entirely about data, and only 2 sentences about the methodology.

Results:is too short.

Discussion:only focused on the similarities and dissimilarities to other previous studies, instead of discussing about the findings. Many parts could be given in the Results Section.

Conclusions:is mainly discussion, and not conclusions supported by the results.

Major comments

1.Figure 1: the legend and ticks are in Italian. Although the English words corresponding to the legend are given in the caption, it seems that the authors just got this figure from a report, instead of making it.

2.Section 2.1, Lines 147-180: there is no any reference and/or analyses to support this section.

3.Figure 2: It has seemingly been taken from the internet. I tried to google it and found the exact map. In this case, the authors need to consider copyright or make the map by themselves.

4.Lines 183-189: It can be removed!

5.Table 1: It is like a snapshot from a pdf file. The author did not even mention the unit of coordinates.

6.Figure 3: again, a copied map (Regions of Italy) from the internet, but without mentioning the copyright. Very low quality, and impossible to see the locations and name event at 190% zoom.

7.Table 2: it is like two photos have been attached together from a source. If you follow the columns, it is much more visible.

8.Figure 4: what does it want to show? Why is it important to be included in the manuscript? Why Max and min temperature that do not have significant effects on Swabs+%?

9.Line 263 “the scientific literature”: please give some references as examples.

10.The results are almost giving no direction. Not any scatter plot, even without the significant correlations. Not any figure showing the regression line on the data!!!

11.Line 272 “All levels of significance ...”: when it is significant at p>0.01, for sure it is significant at p<0.05 and so on. No need to write like this.

12.Lines 317-319: Is there any reference for this? Or did the authors do some analyses regarding that?

13.Lines 320-325: this is only one ling sentence. It is very difficult to understand what the authors want to say. It must be shortened.

14.Figure 5: What is FPS? Is it First Positive Swabs? It must be mentioned. What is the turquoise colour in the legend? The dates are in Italian.

15.Lines 390-392: This conclusion is not supported by your results that showed no correlations with temperatures.

16.Lines 397-418: are mainly discussion because you have not done any of those analyses, albeit if there are references for these sentences.

17.Lines 436-438: How did you calculate the population density is 2200 inhabitants/km2for Brescia and 430 inhabitants/km2for Milan. Based on the Table 3, these numbers are totally wrong.

18.Lines 443-446: It is related to the Discussion Section as it is not based on your results.

Minor comments

The manuscript needs a professions language proofreading before re-submission. So, I did not go through all of such issues; but for example:

1.Line 202: “AWT” to “AWS”

2.Figure 4, caption: “march” to “March”, “avril” to “April”

3.Lines 297-298: it seems that these two paragraphs must be attached to each other.

4.Line 357: “tab 3” to “Table 3”

5.Table 3, and in the most parts of text: “Province” to “city”

Sincerely,

Round 2

Reviewer 1 Report

Thanks for your reply. 

Reviewer 2 Report

Thank you for correcting article. The article is suitable for printing.

Reviewer 3 Report

Dear authors,

Thanks for revising your manuscript and satisfactorily replying to my comments. I have no further comments.

Sincerely,